# Noninvasive Glucose Sensing In Vivo

**DOI:** 10.3390/s23167057

**Published:** 2023-08-09

**Authors:** Ho Man Colman Leung, Gregory P. Forlenza, Temiloluwa O. Prioleau, Xia Zhou

**Affiliations:** 1Department of Computer Science, Columbia University, New York, NY 10027, USA; xia@cs.columbia.edu; 2Barbara Davis Center for Diabetes, University of Colorado Anschutz Medical Campus, Aurora, CO 80045, USA; gregory.forlenza@cuanschutz.edu; 3Department of Computer Science, Dartmouth College, Hanover, NH 03755, USA; temiloluwa.o.prioleau@dartmouth.edu

**Keywords:** noninvasive glucose sensing, diabetes, interstitial fluid

## Abstract

Blood glucose monitoring is an essential aspect of disease management for individuals with diabetes. Unfortunately, traditional methods require collecting a blood sample and thus are invasive and inconvenient. Recent developments in minimally invasive continuous glucose monitors have provided a more convenient alternative for people with diabetes to track their glucose levels 24/7. Despite this progress, many challenges remain to establish a noninvasive monitoring technique that works accurately and reliably in the wild. This review encompasses the current advancements in noninvasive glucose sensing technology in vivo, delves into the common challenges faced by these systems, and offers an insightful outlook on existing and future solutions.

## 1. Introduction

Diabetes affects 37.3 million people in the U.S. and the number is projected to increase [1]. People with diabetes (PwD) have abnormal levels of insulin in their bodies, causing their glucose levels to fluctuate uncontrollably. Failure to manage blood glucose within a healthy range would damage blood vessels and further cause organ failures, leading to complications such as heart disease, kidney disease, and stroke [2]. Therefore, it is vital for PwD to monitor and regulate their blood glucose level.

Currently, over 48% of people with type 1 diabetes (T1D) in the U.S. utilize continuous glucose monitors (CGMs) to closely monitor their glucose level [3]. In essence, the CGM system works by inserting a flexible micro-needle beneath the skin, which measures the glucose concentration within the body fluid known as the interstitial fluid (ISF). Previous research has demonstrated a strong correlation between the ISF glucose level and the blood glucose level [4,5], making CGMs an effective tool for individuals with T1D to manage their glucose levels. Sensing glucose concentration in ISF does not require direct access to the bloodstream and thus eliminates the inconvenience of frequent finger prickings. However, the insertion of a micro-needle can still cause discomfort and potential bacterial infection [6], highlighting the need for a noninvasive approach to improve the overall experience of glucose management.

Noninvasive glucose sensing has been an active research topic in recent decades. To assess the rationale and to demonstrate the feasibility of existing techniques, many studies have used glucose solutions [7,8,9], synthetic tissues [10,11,12,13], and small animals [14,15,16,17]. While these methods provide a good starting point, it is crucial to evaluate their performance in real-world scenarios with actual users, particularly PwD.

This paper presents a comprehensive review of the current advancement in noninvasive glucose sensing technologies in vivo. While there are a few similar reviews on this topic [18,19,20,21], our focus here is specifically on the in vivo studies that demonstrate the feasibility of these technologies for human use. The paper is structured as follows: we will start by summarizing the basics of diabetes, glucose management, different types of biological fluids, and evaluation metrics used by diabetes communities in Section 2. Then, we will discuss various noninvasive sensing techniques in Section 3, where these techniques are categorized based on the sensing modality. Finally, we will discuss common barriers associated with these techniques in Section 4 and provide expert opinions on possible solutions to these barriers in Section 5.

## 2. Background

### 2.1. Diabetes

T1D is a chronic disease in which the pancreas produces little to no insulin. This is caused by an autoimmune reaction where the body attacks and destroys the pancreatic B-cells that are responsible for producing insulin [22]. T1D is the most common form of diabetes in the pediatric population, and incidence rates continue to rise [23,24]. Usually, when food is being digested, carbohydrates are broken down into glucose and absorbed into the bloodstream. This increase in blood glucose level causes the release of insulin from the pancreas. Here, insulin plays a crucial role in moving glucose from the blood into the cells for energy production. This lowers the amount of glucose inside the blood and maintains the glucose level in a healthy range. However, without sufficient insulin, glucose accumulates in the bloodstream resulting in elevated blood glucose levels (hyperglycemia).

Chronic hyperglycemia is damaging to the vessels that supply blood to vital organs. It leads to microvascular complications including blindness, kidney failure, peripheral nerve dysfunction, as well as macrovascular complications such as heart disease and stroke [2]. The landmark Diabetes Control and Complications Trial (DCCT) [25] showed that intensive insulin treatment could reduce the risk of these complications but at the same time increase the rate of low blood glucose level (hypoglycemia) [26,27], which may lead to acute complications such as seizures and cognitive impairment [28]. Since there is no cure for diabetes currently, it is crucial for PwD to properly manage their blood glucose levels.

### 2.2. Glucose Management

Effective management of diabetes requires intensive monitoring of blood glucose levels and exogenous use of insulin as necessary. The main goal of T1D treatment is to maintain the glucose level within the target range, which is generally considered to be 70–180 mg/dL [29]. The amount of time the glucose level is in the target range per day is known as the time in range (TIR). People with T1D should aim to reach TIR > 70%, which is shown to reduce the risk of long-term complications [30].

Assessments of blood glucose have traditionally been obtained via a blood sample obtained by pricking the fingertip. The sample is then analyzed with a glucometer using a test strip [31]. The test strip contains a glucose oxidase enzyme that reacts to the glucose molecules in the blood sample and produces an electric current. The current is then picked up by a glucometer and converted into a readable glucose level. This method has been more accurate and effective than other self-monitoring approaches, though it is also highly invasive and uncomfortable for PwD.

Alternatively, PwD can monitor their blood glucose levels using a continuous glucose monitor (CGM), which reports glucose level every 1–5 min [32]. CGM is based on the same working principle as a glucometer. Rather than measuring the blood glucose level directly, a thin and flexible sensor is inserted under the skin and measures the glucose concentration in the interstitial fluid (ISF) instead. It has been shown that the ISF glucose level is highly correlated to blood glucose level [4,5] but with an average lag time of 8–10 min [33]. However, CGMs have a relatively low adoption rate because of the inconvenience of inserting the sensor into the skin, the physical and psychological burden of having a visible device on the body, and the high financial burden of replacing the biosensors regularly (every 7–14 days) [34]. Noninvasive glucose sensing is essential for broader adoption of real-time glucose assessments, which can lead to better outcomes for PwD.

### 2.3. Biological Fluid

The goal of noninvasive glucose sensing is to measure the blood glucose level without direct access to the bloodstream. This poses a huge challenge as blood vessels are usually hard to reach. They are located below the epidermis, which includes skin cells, proteins, pigments, and more. Hence, prior studies have looked into alternative biological fluids with similar glucose dynamics as blood glucose. The ISF layer is the most studied fluid type for glucose sensing.

ISF is a layer of fluid that fills the space between cells. It acts as a buffer that brings oxygen and nutrients from the blood capillaries to the cells and waste products back to the blood. This leads to a high correlation between blood glucose and ISF glucose [4,5]. Additionally, ISF is close to the skin surface, making it an easier target than blood. ISF can also be extracted from the skin to the surface via reverse iontophoresis (Section 3.3.1) and analyzed with conventional glucose sensors.

The main limitation of ISF-based glucose sensing is the delay (8–10 min) between blood glucose and ISF glucose, meaning that changes in blood glucose are reflected in ISF glucose after a short amount of time [33]. The reason is that it takes time for glucose in the bloodstream to diffuse into the ISF. This lag becomes longer during hypoglycemia as a result of the body’s stress response. This is particularly undesirable because minute-by-minute blood glucose values would be the most clinically beneficial during hypoglycemia but this is when the lag is greatest. Given this, ISF-based methods have larger errors when taking glucose measurements during the time when the glucose level is changing rapidly, such as right after a meal or right after correcting hypoglycemia [35].

In addition to ISF, other types of biological fluid such as tears [36,37,38,39,40,41], sweat [42,43,44,45,46,47,48], and saliva [49,50,51,52,53] have also been explored for glucose sensing. Overall, they have a slightly longer lag time than ISF (∼10–15 min [54,55,56]), and glucose concentration in these types of fluids is much lower than blood glucose (∼1–2%), making it hard to estimate glucose level accurately [57,58,59,60]. The weak correlation of these biological fluids with blood glucose is also a limiting factor [61,62]. Table 1 provides a summary of different types of biological fluids used in prior literature for glucose sensing.

### 2.4. Evaluation Metrics

In the diabetes research community, researchers and clinicians use various metrics to evaluate the clinical accuracy of glucose sensing systems. The most commonly used one is Clarke’s error grid analysis, proposed by Clarke et al. in 1987 [65]. It is a graph with the predicted glucose level against the reference glucose level, also known as the ground truth (Figure 1a). The error grid is divided into five zones, namely Zones A, B, C, D, and E. Zone A is the region where the predicted value is within 20% deviation from the reference value and is considered clinically accurate. Zone B is outside of Zone A but is still clinically acceptable. This means the predicted glucose value has an error larger than 20% of the actual glucose level, but it will not lead to incorrect treatment decisions. Zone C is the region where a much higher or lower glucose value is predicted, leading to unnecessary treatment. For Zone D, the actual glucose value is either too high or too low while the predicted glucose value is within the healthy range. This fails to detect hyperglycemia or hypoglycemia, which is dangerous to the user. Finally, Zone E will lead to an incorrect decision as the opposite event is reported; a hyperglycemia event will be reported as hypoglycemia and vice versa.

In 2000, an updated error grid known as the consensus error grid was proposed by Parkes et al. [66]. This error grid was based on a survey of 100 endocrinologists and is shown in Figure 1b. Similar to the Clarke error grid, it is divided into five zones but with slightly different definitions. Zone A refers to predictions that are clinically accurate, where they will have no effect on clinical action. Zone B corresponds to altered clinical action in which the error in the predicted glucose value will affect the treatment decision. Zones C, D, and E are similar to Clarke’s error grid, which correspond to overcorrection, failure to correct, and anti-correction. Similarly, a second consensus error grid with relaxed restrictions was designed for people with type 2 diabetes on insulin therapy since this population can potentially tolerate larger errors in the blood glucose value compared with people with T1D (Figure 1c). However, the consensus error grid for type 2 diabetes is not widely used because it has lower clinical performance.

Other than the error grids, the mean absolute relative difference (MARD) is also used to describe the accuracy of glucose monitoring systems. It is the mean percentage error of the predicted glucose value that is straightforward and easy to interpret.
$$\mathrm{MARD}=\frac{1}{n}\sum^{n}\frac{\left|\mathrm{predicted}\;\mathrm{glucose}-\mathrm{reference}\;\mathrm{glucose}\right|}{\mathrm{reference}\;\mathrm{glucose}}$$

Finally, the percentage of agreement rates %15/15, %20/20, and %30/30 are used to relax the requirement of these systems at a low glucose level. Take %15/15 as an example; it describes the percentage of predictions where for each predicted-value–reference-value pair: (i) if the reference value is greater than or equal to 100 mg/dL, then the predicted value is within 15% of the reference value and (ii) if the reference value is smaller than 100 mg/dL, then the absolute difference between the predicted value and the reference value is smaller or equal to 15 mg/dL. This can be generalized into the following:$$\begin{array}{c} x\%/x=\%\left\{\begin{array}{cc} \frac{\left|\mathrm{predicted}\text{-}\mathrm{reference}\right|}{\mathrm{reference}}\leq x\% & \mathrm{if}\;\mathrm{reference}\geq 100\;\mathrm{mg}/\mathrm{dL}\\ |\mathrm{predicted}\text{-}\mathrm{reference}|\leq x & \mathrm{if}\;\mathrm{reference}<100\;\mathrm{mg}/\mathrm{dL} \end{array}\right. \end{array}$$

In 2020, The U.S. Food and Drug Administration (FDA) published an updated standard on Self-Monitoring Blood Glucose Test Systems for Over-the-Counter Use [67]. It requires a glucometer to have 95% of the predicted glucose value to be within 15% of the corresponding reference measurement across the entire claimed measuring range of the glucometer. Additionally, 99% of the predicted glucose value has to be within 20% of the reference measurement.

## 3. Noninvasive Sensing Techniques

Prior studies have proposed various noninvasive glucose sensing techniques. We categorize these existing techniques into six types based on the sensing modality (Figure 2). In particular, optical techniques have been actively studied the most. We further divide them into two classes: (1) direct sensing techniques that measure the glucose’s optical properties to infer the glucose level, and (2) indirect sensing techniques that detect the change in properties of the tissue and blood to infer the glucose level.

### 3.1. Optical Techniques—Direct Sensing

Glucose molecules possess multiple optical properties that can be exploited for glucose sensing. As light penetrates the skin into the layers containing ISF, it interacts with the glucose inside the fluid. By analyzing the relationship between the change in glucose level and the corresponding change in the optical signal, we can directly predict the blood glucose level. In this section, the principles behind the common approaches including infrared absorption spectroscopy, Raman spectroscopy, photoacoustic spectroscopy, polarimetry, and fluorescence are explored.

#### 3.1.1. Infrared Absorption

One of the most extensively studied techniques for noninvasive glucose sensing is the absorption-based approach. When light passes through the glucose molecules, they selectively absorb some wavelength bands depending on the chemical structure. This absorption is proportional to the concentration of glucose, and the glucose level can be determined by observing how much light in those bands is absorbed. Note that for absorption-based approaches performed on human, the infrared (IR) band, particularly the mid-infrared (MIR) and the near-infrared (NIR) spectral region, is generally used. This is because glucose’s molar absorptivity at the visible light and ultraviolet light (UV) region is weak [68]. Additionally, exposure to UV and any other light with a shorter wavelength is damaging to the DNA in cells as they are ionizing.

When IR light passes through a molecule, it causes the covalent bonds inside the molecule to vibrate. The covalent bond is spring-like in nature; it can stretch and bend when energy is applied. It vibrates mostly at the resonant frequency, where the frequency of the light matches the vibrational frequency. This absorbance can be described using the Beer–Lambert Law:A=ϵCL,
where *A* is the absorbance, ϵ is the molar absorption coefficient dependent on the wavelength λ, *C* is the concentration, and *L* is the optical path length. *A* can also be expressed as transmittance such that $A=\log(\frac{I_{0}}{I})$ where $I_{0}$ is the incident intensity and *I* is the transmitted intensity. This shows that given a light source of fixed wavelength and a constant optical path length, the amount of light absorbed is directly proportional to the concentration of a substance. Note that this only tells the presence of a certain chemical structure rather than a molecule. Hence, it is important to focus on absorption wavelengths that are relatively unique to glucose. Figure 3 shows the principle of sensing glucose using the infrared absorption property.

In the MIR region (4000–400 cm${}^{-1}$), glucose absorption peaks in this region are associated with the stretching and bending of C-C, C-H, and O-H bonds. Note that historically, MIR is described in the unit of wavenumber (cm${}^{-1}$) and NIR in terms of wavelength (nm). The strongest band within this region is at 3550 cm${}^{-1}$ which is the fundamental frequency for the stretching of the O-H bond and 2961 cm${}^{-1}$ and 2947 cm${}^{-1}$ for the stretching of the C-H bond [69]. The region between 1200 cm${}^{-1}$ and 800 cm${}^{-1}$ is known as the fingerprint region, which contains multiple bands corresponding to C-H bending and C-O-H bending. Glucose has sharp absorption bands in this region, which are more distinguishable from other substances also present in the skin (e.g., water and fat) [70]. Additionally, MIR light scatters less on the skin compared with NIR [71]. The poor penetration depth (∼100 μm) of MIR due to the strong absorption of water is the major drawback [72,73,74]. This means that 63% of the light is absorbed by the tissue after it penetrates the skin for 100 μm.

In the NIR region (700–2500 nm), the main absorption peak is located at 1600 nm which corresponds to the first overtone of the C-H stretch. There are other absorption peaks such as 960 nm (O-H second overtone), 1150 nm (C-H second overtone), 1450 nm (O-H first overtone), and 1900 nm (combinations of the O-H stretch and H-O-H bending of water) [75,76]. However, they are masked by other substances also present in the skin, such as water, lipids, red blood cells, proteins, and acids [69,77], making them more difficult to analyze. The absorption coefficient of glucose with respect to NIR is also weaker compared with MIR because they correspond to overtones. Conversely, its higher penetration depth of up to 5 mm [78] is the main advantage over MIR light.

Numerous attempts have been made to sense glucose noninvasively based on IR absorption in persons with diabetes [79,80,81,82] and without diabetes [83,84,85,86,87,88,89] in recent decades. Malin et al. [80] demonstrated the feasibility of sensing glucose with NIR diffuse reflectance spectroscopy over the 1050–2450 nm wavelength range with seven participants. They found that there are many factors that cause variations in tissue sampling, such as the roughness and hydration of the skin surface, tissue displacement caused by contact pressure, and skin temperature, which have to be considered to accurately predict glucose level. Lam et al. [84] made a similar observation, where the effect of physiological influence is also one of the sources of error. Both concluded that a calibration model developed for each participant using partial least square regression can potentially solve the problem.

Since the skin condition affects system performance, Burmeister et al. [83] evaluated the performance of NIR spectroscopy for predicting glucose level at six body sites, i.e., the cheek, lower lip, upper lip, nasal septum, tongue, and webbing tissue between the thumb and forefinger. They found that glucose has overlapping peaks with water and fat and that the tissue with the lowest fat will produce the least amount of noise. The tongue has the least fat while the webbing has the most, which means measurement of the tongue is preferred. Chen et al. [88] then studied the effect of finger contact pressure on the performance of glucose sensing using MIR spectroscopy. The performance was improved using firm finger pressure. It is suggested that applying pressure on the finger could flatten the intertwined interface of the epidermis, which brings the dermis layer closer to the surface so that more ISF can be sensed with the same penetration depth.

Instead of using a full IR spectroscopy setup, which is expensive and bulky, techniques to minimize the sensing hardware have also been actively researched. Liakat et al. [85] proposed the use of a hollow core fiber to deliver light from an external cavity QC laser. The backscattered light from the skin is then captured using a bundle of six fibers and analyzed with a commercial liquid nitrogen cooled mercury cadmium telluride detector. They verified the setup with three participants without diabetes, where 84% of glucose predictions fall into Zone A of the Clarke grid and the rest in Zone B. Haxha et al. [86] further miniaturized the sensor to a wearable form factor. The proposed prototype measures the transmittance of a single NIR diode of 940 nm across the finger. Similarly, Kasahara et al. [87] looked at the most prominent wavenumber features in the MIR region that could lead to the best performance, which are 1050 cm${}^{-1}$, 1070 cm${}^{-1}$, and 1100 cm${}^{-1}$. Both evaluated the performance of their choices with a small number of persons without diabetes. While they showed good performances using only a few wavelengths, more experiments are needed to verify the selection of wavelengths on PwD who have a large variation in glucose value and people without various skin tones and conditions.

Deep neural networks can also be utilized to further improve the performance of these sensing systems. Han et al. [89] improved the performance of the NIR absorption approach by combining the partial least square regression approach with a stacked auto-encoder (SAE) deep neural network. They suggested that the addition of SAE can better suppress the influence caused by individual differences. The experiment with 19 participants without diabetes successfully predicted the blood glucose level with 97.96% predictions in Zone A of the Clarke error grid. Srichan et al. [79] trained a shallow dense neural network to predict blood glucose level using NIR data and personalized medical features that include gender, age, weight, height, and blood pressure. Using a training dataset of 401 participants, they achieved a 96.6% accuracy on a testing dataset of 234 individuals, with all predictions falling within Zone A of the Clarke error grid. These results showed that the use of neural networks can potentially improve the performance of these sensing systems.

Companies have been developing prototypes based on IR spectroscopy. An Israeli company developed OrSense in 2007, a noninvasive glucose sensor using NIR spectroscopy [90]. The proprietary technique used is termed occlusion spectroscopy, where two NIR spectra are obtained: one normally at a finger using a ring-shaped device and the other one when the device applies an over-systolic pressure to the finger to occlude the blood flow. They claim that this creates a new blood dynamic where the unique signal can be analyzed to infer the glucose concentration in the blood. The system was evaluated with 12 persons with T1D and 11 with Type 2 Diabetes. They achieved a 17.2% MARD with 69.7% predictions in Zone A and 25.7% in Zone B. Even though they obtained CE approval, it has never been commercialized. Another Israeli company developed TensorTip Combo Glucometer in 2018, which is commercially available [91]. The system uses four monochromatic light sources in the visual to IR spectrum (625 nm, 740 nm, 850 nm, and 940 nm) and measures the amount of light traversing across the fingertip to the image sensor. Due to individual differences, the system requires 25 calibrations measured at different hours of the day for several days so that a personal reference model can be generated for glucose prediction. A clinical study [92] has been performed with 36 participants, which included both people with and without diabetes. They achieved a 14.4% MARD with all of the predictions falling in Zones A and B of the consensus error grid. The performance is one of the best among all the other commercialized products, but the performance has to be further improved to reach FDA standards. The large number of calibrations required for the device to function accurately is another drawback of this device.

To summarize, IR spectroscopy is a promising candidate for detecting glucose noninvasively due to the preferential absorption of infrared light by the glucose molecules. It has been demonstrated that an IR spectrum contains rich information to predict glucose concentration. However, the selection of wavelengths remains a challenging problem. NIR has a deep penetration depth in the skin; thus, it also interacts with many interfering molecules that could mask the glucose signal. Conversely, MIR has a poor penetration depth owing to the strong water absorption, but the unique absorption peaks in the spectrum are more distinguishable from other coexisting ISF components. In addition, the NIR laser is a more cost-effective option, but the recent development of quantum cascade lasers (QCLs) has also substantially reduced the cost of the MIR laser. Another obstacle faced by IR spectroscopy is the scattering of light on the skin, which is dependent on conditions such as blood flow, hydration, temperature, and the skin’s subsurface structure. There could be huge variations between individuals and testing sites, which may require additional calibrations.

#### 3.1.2. Photoacoustic Spectroscopy

Photoacoustic spectroscopy (PAS) combines two modalities for sensing: optical absorption and ultrasound detection. The photoacoustic (PA) effect occurs when light is absorbed by a molecule and converted into heat energy, causing an expansion in the tissue. This causes an increase in pressure and subsequently produces acoustic waves that propagate to the tissue surface, which can be detected with an acoustic sensor [93]. The intensity of the acoustic wave is proportional to the amount of energy absorbed [94].

Similar to IR spectroscopy described previously in Section 3.1.1, IR light is used in PAS glucose sensing. However, rather than measuring the light absorbed by the tissue, the intensity of the acoustic wave generated due to the PA effect is detected instead. This offers a few additional advantages over IR spectroscopy: (1) it overcomes the problem of light scattering and reflection since it is only sensing acoustically, which is particularly useful when probing the skin as the skin is highly scattered; (2) the sensing performance is limited by thermal noise only, which is lower compared with the noise present in optical approaches [95]; and (3) the attenuation of acoustic waves by water is weaker, which further increases the penetration depth and the amount of ISF being probed. Figure 4 shows the principle of sensing glucose using the photoacoustic effect.

Various attempts have been made throughout the recent decades [70,96,97,98,99]. For example, Pleitez et al. [100,101] used an external cavity quantum cascade laser (EC-QCL) to rapidly scan the skin in the mid-infrared fingerprint region and to measure the PA signal with a windowless resonant cell. A study with one person with diabetes and one person without diabetes showed promising results, with most predictions falling in Zone A of Clarke’s error grid and the rest in Zone B.

It is noted that the use of EC-QCL coupled with the resonant cell allows for a high signal-to-noise ratio and fast spectra acquisition. However, the skin condition is a major concern since factors like the thickness of the stratum corneum (outermost layer of the epidermis) could substantially affect the amplitude of the signal. Therefore, a testing site with a thin stratum corneum is crucial to the performance of a PAS system. Bauer et al. [102] looked into four body parts, i.e., the thumb, the index finger, the palm, and the arm and determined the optimal location for sensing. A study with one person with diabetes and four persons without diabetes verifies that the thumb is found to be the optimal site due to the thin stratum corneum, minimal fat layer, and good blood circulation. The index finger was next, with a similar performance. Another factor that affects the performance is the PA signals produced by confounding substances that may mask the glucose signal. Sim et al. [103] studied how sweat and sebum components present on the hand could affect the performance of predicting glucose. A 2D position image of the fingertip was acquired, and the locations without any eccrine sweat glands were identified. A spectrum in the mid-infrared region was obtained at these non-secreting locations and was analyzed with partial least square regression. Five experiments with one person with diabetes and another without diabetes resulted in 14.4% MARD with 70% of the prediction in Zone A of Clarke’s error grid and the remaining 30% in Zone B.

PAS can generally penetrate deeper than pure IR spectroscopy approaches. Since the water absorbs less acoustic signals than optical signals, this allows acoustic waves generated at a deeper depth of the skin to propagate to the surface for detection. Additionally, it is unaffected by the scattering of light on the surface as it does not affect the acoustic sensor. However, the use of optics and acoustics results in a relatively complex setup, which could be one of the main reasons why there is currently no announced commercial system that uses PAS to sense glucose level.

#### 3.1.3. Raman Spectroscopy

The approaches introduced earlier are based on light absorption, where the amplitude of light changes after interacting with glucose; for Raman spectroscopy, the change in wavelength is observed instead. Normally, when light interacts with molecules, the photons are absorbed by the molecules, which cause the molecules to vibrate, and then the photons are immediately re-emitted at the same wavelength. This is called elastic or Rayleigh scattering. On some rare occasions, the molecule gains vibrational energy and is promoted to a virtual excited state, while the photon loses energy and is scattered at a lower energy level, causing a shift in wavelength. This is known as inelastic or Raman scattering, and it typically occurs in a small fraction of the incident light, roughly 10 in a million photons [104]. By filtering out the Rayleigh scattered light and analyzing only the Raman scattered light using a spectrometer, bands of light at the lower energy levels with longer wavelengths than the incident light can be observed, which are called Stokes. Conversely, a molecule that is already in a virtual excited state can gain energy from a photon but then decay back to the ground energy state. In this case, the photon will be re-emitted at a higher energy level with a shorter wavelength and the resulting bands on the spectrum are called anti-Stokes. In terms of quantum mechanics, Stokes and anti-Stokes have the same probability to occur, but since there are more molecules in the ground state than the excited states, the Stokes will have higher peaks than the anti-Stokes. Hence, the Stokes have a stronger signal and are usually measured in Raman spectroscopy.

The wavelengths of the bands generated from Raman scattering correspond to the differences in the vibrational energy levels. This is determined using the chemical structure of the molecule, and therefore, the Raman spectrum is typically unique to the molecule. For glucose molecules, the peaks in the Raman spectrum can be found at 911, 1060, and 1125 cm${}^{-1}$ [17,105]. Figure 5 shows the principle of sensing glucose with Raman spectroscopy.

There are a few studies that have explored the use of Raman spectroscopy to predict blood glucose level noninvasively [17,105,106,107,108,109,110]. Ebejder et al. [108] first demonstrated the feasibility of measuring glucose transcutaneously with Raman spectroscopy. They found that the dominant features in the Raman spectrum are mainly collagen I, the main component of the dermis, and triolein, which can be found in subcutaneous fat. These features vary a lot among the participants, suggesting a large variation in chemical compositions. Since these spectral features are relatively distinct from the peaks of glucose [111], this allows for the change in glucose concentration to be observed and measured. Using an excitation light of 830 nm, a study with 17 participants without diabetes was performed and 461 Raman spectra were collected. The spectra data were analyzed with partial least squares regression, and a 7.8% MARD was achieved. A universal calibration is crucial to significantly reduce the number of calibrations performed by the users. Lipson et al. [109] looked at the requirements for a universal calibration in Raman spectroscopy such that a priori information of a person or ground truth of the blood glucose level are not necessary. They first assumed that the Raman spectrum obtained from the skin contains signals from blood, ISF, and intracellular fluid and that the proportions could be site dependent and then formulated an optimization with the volumes of the fluids as the variable. To conclude, a dataset that includes 30 persons with spectra collected from more than 300 separate skin sites is sufficient to provide a universal calibration. Li et al. [106] then noticed the poor penetration depth of the excitation light and proposed sensing the glucose in the microvessels of the nailfold, which is the small area of the skin beneath the nail. This allowed them to probe at the blood glucose directly. Experiments with 12 people without diabetes resulted in all predicted values falling within the clinically acceptable region.

In 2018, a prototype called GlucoBeam that uses Raman spectroscopy to sense glucose concentration was developed and is in the process of commercialization [107,110]. They performed a comprehensive clinical study with PwD who experience a much larger range of glucose levels. With a cohort of 15 PwD and a duration of 28 calendar days for each person, 94% of the predicted glucose values fell in the clinically acceptable zones, with the remaining values falling within Zone C (5.8%) and Zone D (0.5%).

The problems affecting Raman spectroscopy are similar to those of IR spectroscopy. It has a limited penetration depth and is affected by physiological effects and environmental noises. Additionally, the Raman signal is very weak due to its low-occurrence nature. On the contrary, it is less affected by water compared with IR spectroscopy since water is a weak Raman scatterer. Additionally, Raman systems are robust against the scattering of light on the skin since the wavelength of the incident light is filtered at the detector.

#### 3.1.4. Polarimetry

Glucose is optically active, meaning the direction of linearly polarized light will be rotated when it passes through a glucose solution. Glucose has 4 chiral centers and 16 stereoisomers, i.e., molecules with the same composition but different spatial configurations. A pair of stereoisomers that are a mirror image of each other but non-superposable are called enantiomers, which rotate the plane of polarization by the same amount but in opposite directions. If an equal amount of the two enantiomers are mixed together, a racemic mixture is formed and it is optically inactive. However, the naturally occurring form of glucose is called d-glucose, or dextrose, which is one of the enantiomers that rotates light (589 nm) clockwise by +52.7° dm${}^{-1}$ (g/mL)${}^{-1}$ at 20 °C [112]. The other enantiomer, i.e., l-glucose, is not naturally occurring but can be synthesized in a lab. It is generally accepted that l-glucose is not found in living organisms, but Stefan et al. [113] detected a negligible amount in human blood. The amount of rotation can be represented by the following equation:α=R(λ,T)·C·〈L〉
where α is the measured optical rotation, R(λ,T) is the rotary power of with respect to wavelength λ and temperature *T*, *C* is the concentration of glucose, and 〈L〉 is the optical path length [114]. If the wavelength, temperature, and optical path length are kept constant, then the measured optical rotation is directly proportional to the concentration of glucose in the sample. Figure 6 shows the principle of sensing glucose with polarimetry.

Prior studies have extensively studied the use of polarimetry to measure glucose in an aqueous solution [7,115,116,117,118,119]. These studies reported a high correlation between the light signal and the glucose concentration. Aqueous humor, a liquid found inside the anterior and posterior chambers of the eye, has also been studied with promising results [120,121,122]. Experiments performed on rabbits have demonstrated the high accuracy of the method, but the main obstacles are the change in corneal birefringence caused by motion artifacts which may mask the signature of glucose, and the difficulty of mounting the sensing equipment on the eye still remains unsolved. To sense the glucose in ISF, the real challenge lies in handling the scattering of light at the skin. Li et al. [123] sensed the ISF glucose by measuring the change in light intensity when polarized light is rotated by glucose molecules. Linearly polarized light is projected onto the skin, which after interacting with the glucose in the ISF layer, is reflected onto a second polarizer. The amount of light passing through the whole setup is proportional to the concentration of glucose. To remove the depolarized light caused by the scattering, two orthogonal linearly polarized lights were used to cancel out the effect. With 41 participants including people with and without diabetes, they achieved a performance of 10% MARD with 89% in Zone A and 11% in Zone B of the Clarke error grid.

While glucose is a good optical rotator, the highly scattering properties of skin depolarize most of the light, leaving only a tiny portion of the reflected light still carrying the polarization information [124]. Additionally, the poor selectivity of this technique due to the abundance of other optical rotators in the skin (e.g., albumin and collagen) is another problem that has to be solved. This explains why polarimetry generally does not work well with the skin.

#### 3.1.5. Fluorescence

A molecule is considered to be fluorescent if it absorbs light of a specific wavelength and re-emits at a different wavelength. The outcome is similar to Raman scattering but it follows a very different physics principle. Glucose itself is not fluorescent but can bind to some proteins and can alter their fluorescence. Specifically, the wavelength of the light re-emitted by these proteins is dependent to the glucose concentration, which can be observed directly as color changes. This requires direct access to the glucose-containing fluid, which is unsuitable for interstitial fluid. Other biological fluids that can be easily collected would be beneficial for this method. In particular, Cui et al. [48] proposed using carbon quantum dot nanomaterials to detect glucose in sweat. However, sweat glucose has a weak correlation with blood glucose and has a very low glucose concentration (see Table 1). Also, the colors can only be classified into several groups (e.g., hypoglycemia, normal, and hyperglycemia) and not an actual glucose level.

#### 3.1.6. Summary

In this section, we look into the various optical techniques that directly sense the optical properties of glucose molecules to infer blood glucose level. We go through the principle of each technique and its related publications. A summary of these techniques is presented in Table 2 and a comparison of their performances is provided in Table 3, highlighting the wavelength of the light source used and the sensing location.

### 3.2. Optical Techniques—Indirect Sensing

Direct measurement of the properties of glucose inside the skin is difficult; hence, researchers look into the effects of glucose on the tissues and blood. These properties are usually easier to sense, but they have a weaker correlation with blood glucose since many other factors, including physiological and environmental factors, can also affect them. The optical techniques including Surface Plasmon Resonance (SPR), Optical Coherence Tomography (OCT), and Photoplethysmography (PPG) are discussed in this section.

**Surface Plasmon Resonance.** This phenomenon occurs when the electrons on a metal surface are excited by an incident light at a certain angle and then propagated along the surface. The angle that triggers SPR depends on the analyte’s refractive index at the metal surface. Since the amount of glucose in the ISF changes its refractive index, we can measure the refractive index of the tissue to predict glucose concentration. This has been studied in [125,126] to sense glucose solution, but no work has been carried out with human skin. Aside from the refractive index being easily affected by many factors, such as skin hydration, temperature, and sweating, the insensitivity towards small glucose concentrations is the main culprit.

**Optical Coherence Tomography.** This technique detects the change in the scattering coefficient caused by glucose to predict glucose concentration. The change in scattering coefficient of the tissue is caused by a mismatch of the refractive index between the ISF and the membranes of the cells in the tissue, which affects light scattering [127]. To sense the change in scattering coefficient, coherent light is split into two beams with one directed to the skin. The reflected light from the skin is combined with the other beam to produce an interference pattern. The interference produced is then proportional to the glucose concentration. This has been explored in previous studies [128,129,130,131,132,133], but similar to SPR, the refractive index and the scattering coefficient are affected by many other factors.

A modified version of OCT is ultrasound-modulated optical tomography (UOT). In UOT, an additional ultrasound is focused on the sensing region. The ultrasound will change the refractive index and displace the scatterer in the target region. This can add an extra dimension to the data and potentially improve the performance. However, this has only been tested with a vessel-mimicking phantom [134] and achieved a 26.6% MARD.

**Photoplethysmography.** This technique involves predicting blood glucose levels from the PPG signal [135,136,137,138,139,140]. It has attracted increasing attention in recent years due to its low cost and wide adoption in smart devices [141]. A PPG sensor uses light to measure the volumetric changes in the blood inside the skin. A common application of a PPG sensor is to measure heart rate. It is proposed that PPG can also be used to measure glucose concentration. The high-level idea is that more glucose inside the blood can cause resistance in the blood flow. The increased blood viscosity is then reflected in the PPG signal. Typically, the PPG signals are analyzed with a deep learning model, and hence, the correlation between PPG and blood glucose is not well understood. Figure 7 shows the principle of sensing glucose with PPG.

### 3.3. Transdermal Techniques

Rather than sensing glucose through the skin, the transdermal techniques enhance the permeability of the skin so that the ISF can be extracted to analyze directly. Then, the extracted ISF is stored in a medium such as a hydrogel and the glucose in the extracted ISF can be detected with a conventional glucose sensor, such as an amperometric glucose test strip. Three methods are explored by the researchers, and they are reverse iontophoresis, magnetohydrodynamic, and sonophoresis.

#### 3.3.1. Reverse Iontophoresis

Iontophoresis is the process of applying a small electrical current to help charged and polar compounds move across the skin at rates higher than their passive permeabilities [142]. It is used for electrically controlled drug delivery due to the often polar and charged nature of the drugs compound [143]. On the other hand, the symmetry of iontophoresis means that it can also extract compounds from the skin, which is known as reverse iontophoresis. The two principal mechanisms involved in the process are electromigration and electroosmosis. The former describes the movement of ions across the skin due to the flow of current. With sodium (Na${}^{+}$) and chloride (Cl${}^{-}$) being the major ions in the interstitial fluid, applying a current to the skin will attract the ions to move to the oppositely charged electrodes. As the skin is negatively charged at physiological pH, the majority of the current carriers through the skin are sodium ions [144,145]. Then, this ion concentration gradient creates an osmotic pressure, causing the ISF to move toward the cathode along with other dissolved compounds such as glucose via electroosmosis. This effectively extracts ISF from the skin to the cathode. Finally, the glucose inside the ISF is measured using conventional glucose sensors. Figure 8 shows the principle of sensing glucose using reverse iontophoresis.

One of the earliest FDA-approved products that use this technology is the GlucoWatch Biographer [64,146,147,148]. It employs two pairs of disposable electrodes for ISF extraction and amperometric glucose sensing, which have to be calibrated with a finger prick test and replaced every 12 h. In the original study, it achieved a promising result with over 96% of the predictions falling in the clinically acceptable region of Clarke’s error grid. However, sweating on the skin contributes to errors in these measurement since there is also glucose present in sweat. More importantly, due to the slow ISF extraction process and the low glucose concentration in the extracted ISF, the extraction process has to be performed over a long duration. This fails to detect rapid changes in glucose level which is crucial in glucose monitoring. Moreover, the prolonged electric current applied to the skin has caused discomfort and irritation to many users, which is also the major downside that is leading to the discontinuation of the device.

Many studies tried to solve the skin irritation problem by using a lower current [149,150,151]. Bandodkar et al. [150], in particular, proposed a body-compliant wearable electrochemical device that is printed on an elastic substrate skin and can be applied to the skin similar to a temporary tattoo. Its low cost, flexibility, and small form factor make it an attractive solution for glucose monitoring. A study with two persons without diabetes over a duration of 1 h demonstrates the feasibility of this method. Subsequent improvements have been proposed for this tattoo-based sensor. Since the proposed sensor is designed for a single measurement, De et al. [151] have extended the tattoo-based sensor for continuous monitoring for up to 8 h. Cai et al. [152] used textile enzymatic electrodes to improve the air permeability of the sensor while maintaining its electrochemical properties. Yao et al. [153] combined the ISF extraction and the glucose sensing into a two-electrode design to further reduce the size of the device. Xu et al. [154] improved the sensitivity of the glucose sensor at low concentrations by using a highly conductive hydrogel.

Efforts to miniaturize the device and to improve the accuracy have been made [155,156], but it is still a long way away from approaching commercialization. For instance, including factory calibration of the sensors is greatly beneficial to the user experience as it removes the need to calibrate the sensor with a finger prick test. The extraction current and duration as well as the sensing frequency have to be optimized so that it is comfortable for the user. The noises caused by the presence of substances such as sweat and other environmental factors have to be resolved until transdermal techniques can be used for continuous monitoring.

#### 3.3.2. Magnetohydrodynamic Extraction

Recently, in 2021, a technique to extract ISF by applying magnetic and electric fields has been proposed [157]. In essence, the magnetohydrodynamic (MHD) effect is produced when a magnetic field and an electric field are simultaneously applied to the skin. This will generate a Lorentz force on the interstitial fluid and cause it to flow outward. Using porcine skin, it was demonstrated that a faster extraction with less amount of energy is applied to the skin compared with reverse iontophoresis. The group further produced a prototype for noninvasive glucose sensing by integrating the MHD fluid extract with a custom-built glucose sensor [158]. In vitro experiments with glucose solutions and ex vivo experiments with porcine skin and a subsequent pilot study with persons without diabetes [159] have shown promising prospects for this method.

This extraction technique greatly improves the experience of ISF extraction and solves some of the problems faced by reverse iontophoresis methods, in particular, the long ISF extraction duration and the lowered glucose concentration in the extracted ISF. However, more experiments have to be conducted to verify the effectiveness and reproducibility of this technique.

#### 3.3.3. Sonophoresis

Ultrasound has been used to enhance the permeability of skin for drug delivery. While the exact mechanism is not known, it has been theorized that the permeabilization is caused by the cavitation of the stratum corneum [160]. Focusing the ultrasound waves on the skin causes high pressures and temperatures in the area, which leads to the formation of bubbles within the stratum corneum. These bubbles are then merged into larger bubbles, eventually creating a path that transverse across the stratum corneum.

Sonophoresis has been used mainly for drug delivery, but a few studies have also looked into extracting ISF using this method [161,162]. It can extract ISF glucose faster than reported reverse iontophoresis techniques and without any pain reported by the participants. However, the glucose level measured from the extracted ISF does not correlate well with the blood glucose level, indicating that some form of alteration to the fluid is made during the extraction.

### 3.4. Electrical Technique

The amount of glucose inside the blood affects the electrical properties of the tissue, such as the dielectric properties and the bioimpedance. Several techniques have exploited these properties to sense glucose noninvasively including microwave, millimeter wave, terahertz wave, and bioimpedance spectroscopy.

**Microwave and Millimeter Wave.** Microwaves [163,164,165,166,167,168] and millimeter waves [169,170,171,172] have been proposed for sensing glucose concentration. Due to their relatively long wavelengths (1 mm–1 m) and corresponding low frequencies (300 GHz–300 MHz), these methods have a higher penetration depth than optical methods. They can reach millimeters or even centimeters under the skin [173], where there are abundant blood vessels. This allows direct access to the blood glucose instead of inferring indirectly from ISF glucose. Blood glucose is known to affect the dielectric properties of the blood [174]. Assuming the change in minerals in the blood plasma has minor or no effects on the electrical properties, the glucose concentration and the dielectric properties can be fitted into a Cole–Cole model [175]. Then, the subtle change in dielectric properties of human tissues can be picked up by a microwave sensor and in turn, used to estimate the glucose level. Even though the relationship between blood glucose and the tissue’s dielectric properties is not fully understood [176,177], these techniques remain attractive alternatives to researchers as they are usually easier to implement and are more cost-effective.

**Terahertz Time-Domain Spectroscopy.** Ultrashort electromagnetic waves in the band where infrared and microwave overlaps is used to probe the skin for sensing glucose concentration in the tissue [178,179,180]. The relationship between the absorption coefficient of Terahertz (THz) radiation and the glucose concentration is not well-defined, but one group suggests that the absorption coefficient is inversely proportional to the glucose concentration as a higher glucose concentration result in a relatively lower water content, which is the main component that absorbs THz [181,182]. The strong water absorption in this band also makes this technique very challenging to extract useful information from the skin regarding the glucose concentration.

**Bioimpedance Spectroscopy.** This method measures the changes in the electrical properties of the skin caused by glucose in the tissue. A small alternating current is passed through the tissue at various frequencies, and the corresponding impedances are recorded and analyzed. The principle is that the concentration of glucose will affect the electric conductivity of the blood. The authors of [183,184,185] have explored using this method for sensing glucose, but the insensitivity toward change in glucose levels is the major downside.

### 3.5. Thermal Techniques

The glucose in the blood is diffused into the interstitial fluid and in turn into the cells for metabolism. Therefore, the amount of glucose in the blood is related to the rate of metabolism which can be indirectly observed from the body heat emission using thermal sensors. Below are two methods that look into these properties.

**Metabolic Heat Conformation.** The general idea of this method is that the glucose concentration in the blood is related to the rate of metabolic oxidation of glucose. This can be determined using a thermal sensor to detect metabolic heat generation and an optical sensor for measuring oxygen saturation, blood rate, and other physiological parameters [186,187]. However, the low performance of this method suggests that there are many other factors also affecting the physiological parameters. For example, PwD do not produce insulin, which prevents glucose from entering the cells. Hence, the glucose concentration increases even though the metabolic rate is unchanged and this event cannot be captured with a thermal sensor. When the blood sugar has raised to a certain level (typically 250 mg/dL), the liver starts to break down fat for energy, which only by then heat is generated. Additionally, PwD struggle with body temperature control [188], which means they have different thermodynamics and have to be carefully considered. Figure 9 shows the principle of sensing glucose using metabolic heat conformation.

**Emission Spectroscopy.** Kitazaki et al. [189] demonstrated the possibility of capturing changes in glucose level remotely using a MIR passive spectroscopic imaging technique based on two-dimensional Fourier spectroscopy. The idea is that glucose emits light in the MIR region and has representative peaks at 9.25 and 9.65 μm. Using a spectroscopic imager, a two-dimensional spectroscopic image is taken on the wrist and the back of the hand placed 600 mm away. While this method has many attractive properties such as being easy to set up and supporting remote sensing, it is a fairly new technique and should be studied more closely to verify its feasibility.

### 3.6. Fusion Techniques

Combining multiple techniques could potentially improve the performance of a glucose sensing system. By using multiple techniques at the same time, researchers hope that the techniques can complement each other to achieve better performance. However, the hardware will inevitably be larger in size to accommodate the additional sensing components. Here, we look into existing work that used multiple techniques.

One of the commercially available products called GlucoTrack used ultrasonic, electromagnetic, and thermal techniques to sense glucose concentration at the earlobe [190]. The glucose concentration changes the density and adiabatic compressibility of the tissue, which directly affects the acoustic velocity that can be detected with the ultrasound sensor. The change in glucose concentrations also affects the impedance and the heat generation of the tissue similar to the bioimpedance spectroscopy and the metabolic heat conformation methods mentioned above. Multiple studies [191,192] have shown promising results, with around 22% MARD for people with type 2 diabetes, but the accuracy still needs to be further improved until it can fulfill the requirements set by the FDA.

Nystrom et al. [193] investigated the effectiveness of combining both NIR spectroscopy and bioimpedance spectroscopy for sensing glucose level. The NIR spectroscopy is useful for determining the glucose level while bioimpedance spectroscopy can detect changes to the body composition, which has been shown to be one of the effects caused by diabetes [194]. A study of 34 PwD and 23 people without was conducted, and the results showed that the combined information can be used to classify people into different levels of neurography, but not enough for predicting glucose. Sometime later, Fouad et al. [195] adopted a similar approach and their study with five people without diabetes achieved all predictions falling into Zone A of Clarke’s error grid, suggesting the validity of combining both approaches.

### 3.7. Summary

In this section, we undertake an in-depth exploration of the six types of techniques, which include optical (both direct and indirect sensing), transdermal, electrical, thermal, and fusion methods. We delve into the working principles behind these techniques and highlight the relevant key publications. Each of these techniques is characterized by its own unique properties and design principles, with their respective advantages and disadvantages outlined in Table 4. Additionally, the studies involving three or more participants and are evaluated with the metrics mentioned in Section 2.4 are compared in Table 5.

## 4. Current Barriers to Noninvasive Glucose Sensing

In the previous section, we investigated the techniques proposed by researchers for sensing glucose noninvasively. While they sound promising, there are still many problems that have to be addressed before they can reach the stage of commercialization. Here, we explore the common barriers and obstacles faced by these sensing techniques.

### 4.1. Confounding Factors

The properties of glucose are leveraged in various sensing techniques, but some substances that are present in the tissue and biological fluid also exhibit similar properties. One such property of glucose is the strong light absorption of specific wavelengths, but substances like water, lipid, and protein also have overlapping absorption peaks with glucose. This means that a change in the concentration of these substances can alter the absorption spectrum even though the glucose level remains unchanged, leading to inaccurate glucose predictions. This is a challenge for all optical methods that rely on absorption, including infrared spectroscopy and photoacoustic spectroscopy.

Methods that infer glucose levels from the properties of the tissue and blood are particularly susceptible, as there are even more confounding factors that would also affect these properties. For example, environmental factors including temperature and moisture content are the major source of noise for methods that predict glucose level using the dielectric properties of the skin. This effect is more prominent when the glucose concentration is low. Therefore, to ensure accurate predict glucose levels, the main confounding substances and how to isolate their effects must be understood.

### 4.2. Selection of Sensing Location

The ideal location for glucose sensing should strike a balance between the system performance and the user experience. To maximize performance without compromising usability and convenience, the testing site should be chosen based on its physical and chemical properties. It has been shown that locations with a thin outermost layer of the epidermis, a minimal amount of fat content, and good blood circulation provide the best results for optical methods [83,102].

The outermost layer of the epidermis, i.e., the stratum corneum, is a layer of dead skin cells packed with structural protein keratin and acts as a chemical barrier. As keratin is the primary source that causes scattering in the skin [196], a thin stratum corneum reduces the optical path through the keratin, bringing the ISF closer to the surface. This effectively reduces the scattering and absorption of light when passing through the skin and eventually increases the signal-to-noise (SNR) ratio. As fatty tissues also absorb light, selecting a testing site with less fat content can further improve the SNR. Additionally, a reduction in fat content and better blood circulation enhance the glucose diffusion into the ISF, resulting in a faster equilibration between blood glucose and ISF glucose and minimizing the lag time problem [102].

Multiple sites have been proposed in previous studies, with [83] suggesting the tongue and [102] the fingertips. While the tongue fits the aforementioned requirements, it will result in discomfort when a user has to stick out their tongue and place it on the sensing interface. The fingertip is the preferred choice because of its accessibility, convenience, and compatibility with the requirements for optimal performance. This is particularly crucial for continuous monitoring as the sensor has to be attached to the testing site for an extended period of time. While the performance of the sensing system is important, the success of the technology will ultimately depend on the user’s comfort and convenience.

### 4.3. Glucose Distribution

When the skin is probed with a sensor, we expect a signal from the ISF glucose to be received. In reality, the signal can come from glucose in the blood, ISF, and intracellular fluid (fluid inside the cells). Depending on the testing site and the penetration depth of the excitation light, it is possible that all three fluids are actually probed at the same time [109]. Therefore, the glucose dynamics of all these fluids should be considered and modeled together.

The distribution of glucose varies among the fluids. For instance, ISF glucose lag time depends on the location of the fluid. The lag time during the decay phase can be as quick as 1–3 min in the dermal layer. The model in [197] even predicts a negative lag time for ISF glucose in the epidermis layer, suggesting that the ISF glucose concentration can decrease before the blood glucose level (∼1 min). This phenomenon has been reviewed in [198]. Other properties such as intracellular fluid having a much lower glucose concentration because of the rapid metabolism of glucose [197] and the volume of ISF being generally larger than blood in the skin [199] should also be considered when modeling the data. As the distribution is site-dependent, it is advised that glucose sensing should be performed at the same location consistently.

### 4.4. Model Generalization

One major challenge of noninvasive glucose sensing is the generalizability of the prediction models. These models work well if only one of the variables changes, i.e., glucose concentration, while everything else is kept constant. However, there is a huge difference between individuals; the composition of the body tissues and biological fluid, skin conditions, and physiological state can all vary greatly among people. Some of these differences can be linked to a person’s demographic characteristics. For instance, race could have an effect on the amount of pigment inside the skin, which is a strong light-absorbing substance and would affect absorption-based optical techniques. Males on average have thicker skin than females, which means it is harder to reach the ISF [200]. As a result, less ISF is probed using optical methods or extracted via transdermal methods. The amount of collagen can be inversely correlated with a person’s age as the production of collagen decreases with age [201]. This alters the scattering coefficient of the skin and adds noise to optical methods. The combined effect of all these factors could result in a totally different skin characteristic for each individual.

Underrepresented minority groups, in particular, often suffer from using generalized models as there are limited training data available from these populations. This leads to a lowered sensing performance for these racial and ethnic populations. Personalized models can be utilized to address this issue using models that are tailored to each individual’s specific characteristics. By collecting training data from the user, the model can adapt to the user and better isolate the effect of glucose level on the sensor readings. However, this approach will require a calibration stage that may involve blood glucose measurements from finger prick tests, causing inconvenience to the user. Multiple calibrations may also be necessary from time to time due to changes in the person’s physiological parameters.

### 4.5. Hardware Design

The hardware design heavily affects the system adoption. A machine as large as a washing machine is nowhere attractive to the user, even if it can provide an accurate measurement. Conversely, a wearable device with poor performance is unsafe and unlikely to be approved for clinical purposes. Hence, the key is to balance both the performance and the form factor so that the device is portable and reliable. That said, the accuracy of the glucose monitoring system should be the top priority, even if it means sacrificing compactness. The reason is that there are currently no commercially available products that are accurate enough for clinical usage. As a first step, a portable form factor with a good sensing performance would be ideal. Further miniaturizing of the sensing system should be considered later once a reliable system has been established.

The user’s behavior can have a significant impact on the system’s performance. For example, the pressure exerted by the sensor on the skin can affect the skin’s optical properties. High pressure can cause tissue deformation and displacement of ISF, while low pressure can generate air pockets between the skin and the sensor. In the case of continuous monitoring, additional factors should be taken into consideration. The secretion of sweat and sebum on the skin surface would gradually interfere with the sensor signal, causing a sensor drift over time. Ambient factors such as temperature and humidity all add up as noise to the signal as well. To maintain accuracy under various conditions, it is important to consider and incorporate these effects into the device’s design.

### 4.6. Acquisition of Ground Truth

The accuracy of a glucose monitoring system can be evaluated by comparing its predicted glucose values to a reference value, also known as the ground truth. This reference value is typically collected at the same time the system makes a prediction and is retrieved through an invasive finger prick test and/or a minimally invasive CGM. The reference value serves as the baseline, and the performance of the system can be evaluated by performing a pairwise comparison between the predicted glucose value and the reference glucose value and presenting the result using the evaluation metrics mentioned in Section 2.4.

However, the reference values obtained through the finger prick tests of CGMs can deviate from the actual ground truth by a few margins [202,203,204]. Studies have shown that CGM readings can have an average error of 10%. This error can make it difficult to prove the performance of the system, particularly if time and cost are also considered during the evaluation. Despite the error, this is sufficient to verify a proof of concept and to demonstrate the feasibility of a prototype. However, to definitively prove the accuracy and precision of a system, the results should be compared with those obtained from a laboratory test.

### 4.7. Clinical Study

Many of the methods mentioned above only performed experiments on people without diabetes. The blood glucose level of an individual without diabetes is naturally managed to the range between 70 and 140 mg/dL most of the time. Not only is it significantly smaller than what a person with diabetes would experience (50–400 mg/dL), but also by the definition of Clarke’s error grid, a constant prediction of 100 mg/dL (or any value between 70 and 180 mg/dL) for all reference glucose values in the range of 70–240 mg/dL will fall inside the clinically acceptable region. Therefore, it is biased to report a clinical study that has a majority of the results in the clinically acceptable range with people without diabetes and conclusively infers the feasibility of the method. A clinical study with PwD is very much needed before a conclusion can be drawn.

Moreover, it ignores the effect caused by insulin in the biological fluid [205]. For people without diabetes, it is guaranteed that the insulin level will increase with the glucose level; the insulin dynamic is predictable. On the other hand, PwD may have too much or too little insulin in the body at a given time. The body may react differently to the abnormal level of insulin, which could have an impact to the physiological factors. This additional variable is not seen in the population without diabetes.

## 5. Potential Solution and Future Directions

Despite decades of research, noninvasive glucose sensing remains a huge challenge to this day. Different techniques and modalities have been carefully considered by researchers to be the possible candidate to overcome this challenge. To predict blood glucose level noninvasively, various biological fluids have been studied. While sweat and saliva are easy to collect, their low glucose concentration and the weak correlation with blood glucose are the main reasons they cannot be used to achieve high accuracy. ISF, on the other hand, offers a promising alternative. As it is strongly correlated with blood glucose and is present right below the outermost layer of the skin, ISF has emerged as a good candidate for noninvasive glucose sensing.

Since ISF glucose resides in the skin, it is challenging to measure the glucose concentration directly with great accuracy and precision. Many researchers have turned to indirect techniques that measure the properties of the skin and blood that are also affected by glucose concentration. These techniques usually have the advantage of being easier to set up and more cost-effective, with readily available off-the-shelf sensors to detect the properties on the skin surface. However, the correlation between these properties and blood glucose level is generally weak, as many other physiological and environmental factors also have impacts on the properties. While these techniques show promise in controlled environments, their performance in real-world scenarios remains uncertain. On the other hand, transdermal techniques extract the ISF from the skin directly for further analysis, which sounds attractive at first glance as it removes many interfering substances from the analyte. However, the extraction process is slow and it takes a long time to extract a sufficient amount of ISF for analysis. This causes skin irritation to the user and is not capable of detecting rapid changes in the glucose level.

Therefore, techniques based on glucose properties are more concrete as the measurements have direct correlations with the glucose concentration. In particular, the optical properties of glucose are exploited due to its attractive features; the skin is probed with visible or infrared light, which is truly noninvasive, and a well-controlled light beam within the safety limit is generally safe to the user [206,207]. The major downside is that the light has to pass through the outermost layer of the skin to reach the ISF. Since the light will interact with other substances found in the skin and the ISF, it may cause interference with the glucose signal.

**Improving Optical Sensing Performance.** Glucose readings obtained from the skin can be easily influenced by a plethora of factors. All these factors have to be addressed before a device can be commercialized for use in the wild. This is a challenging problem and cannot be easily solved in one go. In this case, a divide-and-conquer approach would be suitable. Certain variables can be fixed or restricted to some extent. Biological variables such as skin thickness and fat content are relatively constant at a given sensing site. A hardware structural design can be developed to help align the body part (e.g., fingertip) with the sensor to minimize the offset between each sensing session. Instead of relying on the user to apply consistent pressure on the sensor, the device can be designed with a mechanism to apply the optimal pressure on the skin every time the sensor is used. By cleaning and drying the skin before using the sensor, which is a very common practice for optical approaches, the impact caused by sweat, sebum, and other contaminants on the sensor can be minimized. Additionally, a personalized model can be utilized so that it can better adapt to the individual differences among the users. This means that the model is created by collecting data from a single person and training a model either from scratch or by fine-tuning a pre-trained model. This allows the model to learn the signal changes with respect to the glucose fluctuations, with most of the other variables kept constant. However, this approach requires collecting reference glucose values through invasive methods such as finger prick tests, which can be inconvenient to the user.

The remaining factors can be handled with a multivariate analysis. Variables like body temperature and skin hydration can interfere with the sensor readings, which may mask the glucose response and have to be isolated. One possible solution is to combine multiple modalities such that these factors generate contrasting signals among the modalities, which can be distinguished by a model. However, the data collected from multiple modalities will have a higher dimension, and hence, a sufficiently large dataset would be necessary to train the model. The hardware will also have to accommodate the various modalities, which could lead to a more complex design that potentially increases the size and cost of the system.

Given the complexity of the problem with numerous variables, collecting a large dataset in the hopes of training a generalized model may not be practical at this stage. Instead, it is more advisable to first study and understand the impact of each factor on the sensor signal. This allows us to rank them in order of significance and then to address them systematically. In the near future, it is expected that personalized models that are calibrated individually will be first used to demonstrate the feasibility and performance of a technique. Upon verification of the performance, a comprehensive clinical study should be conducted next, aiming to construct a generalized model. With a larger dataset, the use of neural networks is expected to enhance the performance further.

**Clinical Evaluation.** Many techniques are evaluated with glucose solutions and synthesized tissues to demonstrate their potential, but it does not reflect their actual performance in vivo. These experiments simplify the problem by eliminating many variables from the equation, making it easier to establish a good correlation between the measured signal and the reference glucose concentration. They are useful for testing the concept or principle behind a method, and a positive outcome can encourage further exploration of the proposed method.

Some methods are then tested with a small group of individuals without diabetes to show the robustness of the system. At this stage, the reference glucose level is obtained through invasive means, usually with finger prick tests or CGMs. This is valid as long as the potential error induced is considered and acknowledged. On the other hand, it should be noted that the blood glucose level of people without diabetes and, by extension, the ISF glucose level, are managed within a healthy range of 70–140 mg/dL. This narrow range makes it easier to obtain a low MARD value and have most of the predictions fall into the clinically acceptable range of Clarke error grid. Thus, caution should be exercised when interpreting results from studies with individuals without diabetes. Moreover, many studies often collect data only for a limited duration, usually within a few hours. This can lead to overestimating the performance of the system as it is not tested against fluctuations in environmental and physiological factors. The system would probably require another round of calibration to work well at another time.

Given the aforementioned problems, a clinical study with PwD spanning across a substantial amount of time is essential to prove the feasibility of the noninvasive techniques. A wide range of glucose values, normally from 50 to 400 mg/dL, should be used to ensure low error rates for PwD. The study period should last several days or even weeks to test the system against fluctuations in environmental and physiological factors. Last but not least, a diverse population with underrepresented minority groups should also be included in the study so that the technique is generalizable across populations.

Numerous efforts have been dedicated to achieving noninvasive glucose sensing, with optical techniques receiving the most attention over the past decades. In vivo clinical studies involving people with and without diabetes have shown promising results. Nevertheless, there are remaining challenges that require research effort to achieving practical deployment, such as handling user diversity and environmental variations. By combining multiple modalities such as NIR spectroscopy and Raman spectroscopy, collecting a comprehensive dataset from clinical studies involving individuals both with and without diabetes, and leveraging deep learning technology for data analysis, the robustness of these systems could be further improved, bringing them closer to meeting clinical standards. Conversely, some non-optical methods have recently emerged, exhibiting some appealing properties compared with optical methods. However, additional clinical studies are necessary to effectively demonstrate their feasibility on individuals.

With the recent advancements in technology, electronics have become smaller, more efficient, and more powerful. Many previously insurmountable challenges have been overcome. Despite the remaining obstacles, we maintain an optimistic outlook for future progress and development in this field.

## 6. Conclusions

The field of noninvasive glucose sensing still has much room for improvement. Although recent technological advancements have brought about great strides in these sensing systems, making them more convenient and user-friendly, barriers to the commercialization of these systems still remain. This includes calibration issues, racial/ethnic disparities, physiological effects, and form factor of the systems. Addressing these barriers will be a complex and ongoing process, but work is on the way to solving these problems. With continuous effort, we expect an improved system and overall better care for PwD.

## Figures and Tables

**Figure 1 sensors-23-07057-f001:**
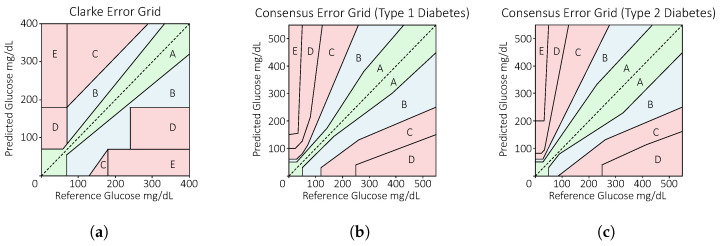
Error grids used for evaluation. (**a**) Clarke error grid; (**b**) consensus error grid for type 1 diabetes; (**c**) consensus error grid for type 2 diabetes.

**Figure 2 sensors-23-07057-f002:**
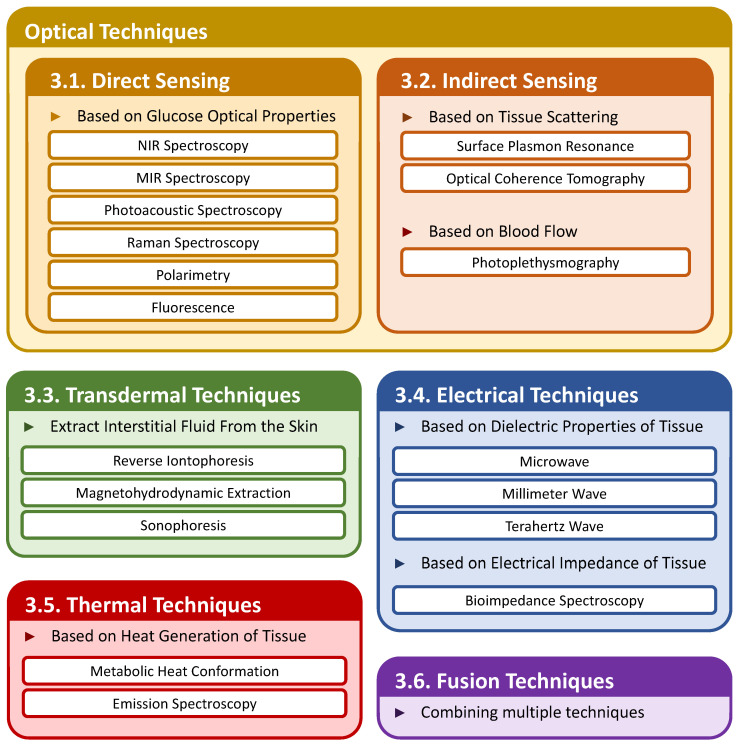
An overview of the sensing methods reviewed in Section 3.

**Figure 3 sensors-23-07057-f003:**
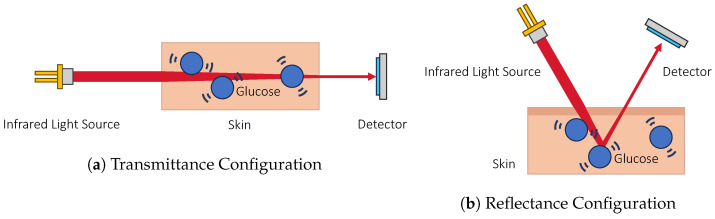
Basic setups for glucose sensing with infrared absorption. Specific bands of infrared light are absorbed by glucose molecules, thereby causing the glucose’s covalent bonds to vibrate. The amount of infrared light absorbed is proportional to the glucose concentration. Two configurations are depicted here: (**a**) IR spectroscopy using the transmittance configuration where the amount of light passing through a sample is measured. (**b**) IR spectroscopy using the reflectance configuration where the light enters the sample, interacts with the glucose molecules, and scatters to the detector.

**Figure 4 sensors-23-07057-f004:**
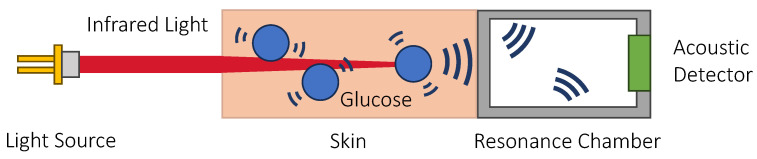
Basic setup for glucose sensing with photoacoustic spectroscopy. When the glucose molecules absorb some specific bands of infrared light and their covalent bonds start to vibrate, acoustic waves are generated, which then propagate to the skin surface and can be captured with an acoustic sensor. The intensity of the acoustic wave is proportional to the amount of light absorbed, which is then proportional to the glucose concentration.

**Figure 5 sensors-23-07057-f005:**
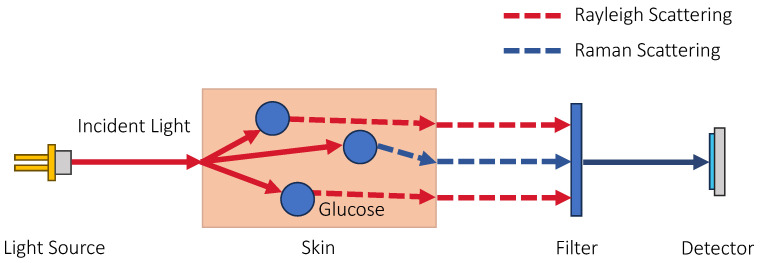
Basic setup for glucose sensing with Raman spectroscopy. When the glucose molecules absorb photons, they vibrate and the photons are re-emitted immediately at the same wavelength, known as Rayleigh scattering. On rare occasions, Raman scattering occurs and the photons are re-emitted at different wavelengths instead. These wavelengths are unique to the molecular chemical structure and can be analyzed by using a filter to block off the incident wavelength.

**Figure 6 sensors-23-07057-f006:**
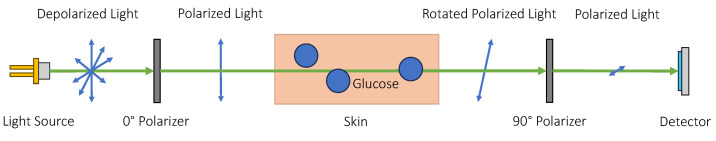
Basic setup for glucose sensing with polarimetry. Light first passes through a polarizer and becomes linearly polarized. The polarized light enters the skin and the plane of polarization is rotated upon interaction with the glucose molecules. The rotated polarized light then leaves the skin and is analyzed to infer glucose concentration.

**Figure 7 sensors-23-07057-f007:**
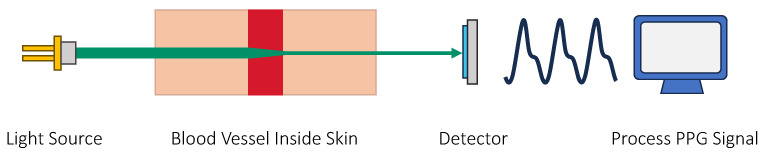
Basic setup for glucose sensing with photoplethysmography. The amount of light absorbed reflects the volume of blood in the skin. Typically, a basic system uses a green or a red light to capture the PPG signal while a pulse oximeter uses a red and an infrared light to additionally deduce the oxygen saturation. The PPG signal is then further processed to infer the glucose sensing, usually with a deep learning model.

**Figure 8 sensors-23-07057-f008:**
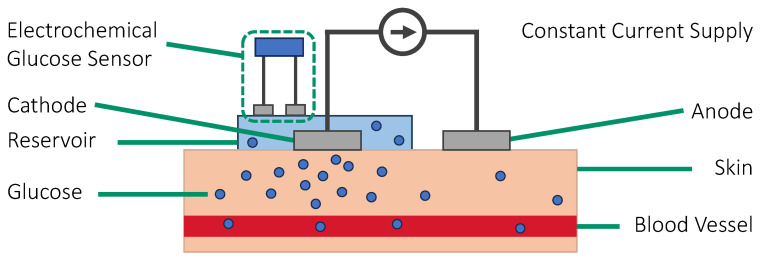
Basic setup for glucose sensing using the reverse iontophoresis technique. A constant current is applied to increase the permeability of the skin and to extract the ISF to a reservoir. The glucose concentration in the extracted fluid is then determined using a conventional electrochemical sensor.

**Figure 9 sensors-23-07057-f009:**
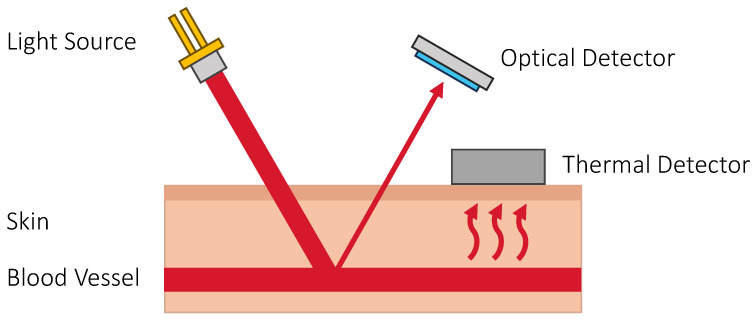
Basic setup for glucose sensing with metabolic heat conformation. Blood flow rate and blood-related information are obtained using the optical sensors, while heat balance and thermal generation information are detected using the thermal sensors.

**Table 1 sensors-23-07057-t001:** A summary of biological fluids including tears, sweat, saliva, ISF, and ISF extracted from the skin to the surface by reverse iontophoresis (RI) are presented. Note that the content in the extracted ISF differs with the ISF inside the skin (Section 3.3.1).

Biological Fluid	Lag Time	Glucose	Advantages	Disadvantages	Maturity
Tears [54,63]	∼15 min	∼2% of blood	Less interfering substances Relatively easy to collect	Low glucose Weak correlation	Low
Sweat [62]	∼10 min	∼2% of blood	Easy to collect	Low glucose Weak correlation Affected by environment	Moderate
Saliva [61]	∼15 min	∼1% of blood	Easy to collect	Low glucose Weak correlation Food particles	High
ISF [4,5]	∼8–10 min	similar to blood	High glucose Good correlation	Hard to reach Interfering substances	High
ISF (RI) [64]	∼15–20 min	∼1% of ISF	Good correlation Relatively easy to collect	Low glucose Skin irritation Affected by sweat	High

**Table 2 sensors-23-07057-t002:** A summary of optical techniques for direct sensing based on optical properties of glucose and their characteristics. The signal-to-noise ratio is used to compare the power of the signal with the background noise when a higher value is desired. The penetration depth describes the distance the light can penetrate, where around 37% (1/e) of the light still remains and is not absorbed by the tissue. The scattering of light on the skin is wavelength-dependent, and the effect is more prominent with shorter wavelengths. Finally, the cost is mostly determined by the light source and the sensor.

Technique	Signal-to-Noise Ratio	Penetration Depth	Affected by Scattering	Cost
Near-Infrared Spectroscopy	Low	Moderate	Moderate	Low
Mid-Infrared Spectroscopy	Moderate	Low	Low	Moderate
Photoacoustic Spectroscopy	NIR—Low MIR—Moderate	NIR—Moderate MIR—Low	None	High
Raman Spectroscopy	High	NIR—Moderate MIR—Low	None	High
Polarimetry	Low	Low	High	Low
Fluorescence	High	None	None	Low

**Table 3 sensors-23-07057-t003:** A summary of optical techniques for direct sensing with in vivo study sorted according to year of publication. The wavenumber of the MIR light is converted to wavelength for easier comparison (10,000,000/wavenumber=wavelength). Participants include people with type 1 diabetes or type 2 diabetes, and people without diabetes. Zones A and B refer to Clarke’s error grid (or the consensus error grid if specified). Other evaluation metrics are also used in some other papers, including correlation coefficient (r), coefficient of determination (R${}^{2}$), mean absolute difference (MAD), and root mean square error (RMSE).

Ref.	Year	Technique	Wavelength nm	Location	Clinical Study			Study Result
					N	w/ Diabetes	w/o Diabetes	
[80]	1999	NIR Spectroscopy	1050–2450	Forearm	7	Yes	No	MARD of 3 participants: 9.1%, 17.6%, 3.6%
[83]	1999	NIR Spectroscopy	630	Multiple	19	n/a	n/a	Tongue is most reliable for glucose sensing
[82]	2002	NIR Spectroscopy	1050–2450	Forearm	9	Yes	No	MARD: 20.6%; Zone A: 63.5%; Zone B: 34.9%
[81]	2003	NIR Spectroscopy	Unspecified	Forearm	1	Yes	No	r: 0.928; standard error of prediction: 32.2 mg/dL
[108]	2005	Raman Spectroscopy	830	Forearm	17	No	Yes	MARD: 7.8% ± 1.8%; R${}^{2}$: 0.83 ± 0.10
[96]	2005	Photoacoustic Spectroscopy	9259, 9381	Forearm	1	No	Yes	A positive correlation is observed
[90]	2007	Occlusion Spectroscopy	10 Unspecified	Finger	23	Yes	No	MARD: 17.2%; Zone A: 69.7%; Zone B: 25.7%
[109]	2009	Raman Spectroscopy	670, 827, 829	Forearm	30	Yes	No	Zone A: 53%; Zone B: 39%; Mean absolute difference: 38 mg/dL
[84]	2010	NIR Spectroscopy	905–1701	Finger	36	No	Yes	r: 0.48; RMSE: 1.34 mmol/l; Zone A + B: 100%
[97]	2012	Photoacoustic Spectroscopy	9225, 9488	Palm	2	No	Yes	r: 0.7. Recommends using 6–10 IR wavelengths
[101]	2013	Photoacoustic Spectroscopy	8197–10,000	Hypothenar	2	Yes	Yes	MAD: 11 mg/dL (without diabetes) and 15 mg/dL (T1D)
[100]	2013	Photoacoustic Spectroscopy	8032–10,000	Hypothenar	1	No	Yes	A windowless PA cell design is proposed and verified
[85]	2014	MIR Spectroscopy	8000–10,000	Palm	3	No	Yes	Zone A: 84%
[98]	2015	Photoacoustic Spectroscopy	905	Palm	30	No	Yes	MARD: 9.61% ± 10.55%. Zone A: 87.24%; Zone B: 12.76%
[86]	2016	NIR Spectroscopy	940	Finger	5	No	Yes	Zone A + B: 100%
[70]	2016	Photoacoustic Spectroscopy	8475, 9259	Finger	n/a	n/a	n/a	R${}^{2}$ = 0.8, uncertainty of ±30 mg/dL at 90% confidence level
[99]	2017	Photoacoustic Spectroscopy	905, 1550	Forefinger	24	No	Yes	MARD: 8.84%; Zone A: 92.86%; Zone B: 7.14%
[91]	2018	NIR Spectroscopy	625, 740, 850, 940	Finger	19	n/a	n/a	Result of 3 Studies: MARD: 17.9%, 14.9%, 17.1%; Zone A + B: 100%, 100%, 98.8% (consensus)
[92]	2018	NIR Spectroscopy	625, 740, 850, 940	Finger	36	Yes	Yes	MARD: 14.4%; Zone A: 96.6%; Zone B: 3.4% (consensus)
[87]	2018	MIR Spectroscopy	1050, 1070, 1100	Finger	6	No	Yes	r: 0.36; Zone A + B: 100%
[110]	2018	Raman Spectroscopy	830	Thenar	35	Yes	No	MARD: 25.8%; Zone A + B: 93% (consensus)
[102]	2018	Photoacoustic Spectroscopy	8032–9852	Multiple	5	Yes	Yes	MAD 16 ± 7 mg/dL. Thumb is most suitable for glucose sensing
[103]	2018	Photoacoustic Spectroscopy	8065–10,526	Finger	2	Yes	Yes	MARD: 14.4% ± 10.5%; Zone A: 70%; Zone B: 30%
[88]	2019	MIR Spectroscopy	6250–12,500	Finger	6	No	Yes	95% certainty and 100% comparability with firm finger pressure
[106]	2019	Raman Spectroscopy	785	Nailfold	12	No	Yes	RMSE = 0.27 mmol/L; R${}^{2}$ = 0.98; Zone A + B: 100%
[123]	2020	Polarimetry	450, 520, 658	Palm	50	Yes	Yes	MARD: 10.0%; Zone A: 89%; Zone B: 11%; r: 0.91; *p* = $1.6\times 10^{-143}$
[89]	2021	NIR Spectroscopy	1050, 1219, 1314, 1409, 1550, 1609	Finger	19	No	Yes	r: 0.92, Zone A: 97.96%
[107]	2021	Raman Spectroscopy	830	Thenar	15	Yes	No	MARD: 26.3% ± 10.8%; Zone A + B: 93.6%
[79]	2022	NIR Spectroscopy	850, 950, 1150	Finger	635	Yes	Yes	Zone A: 100.0%

**Table 4 sensors-23-07057-t004:** A summary of the advantages and disadvantages of the types of techniques used for noninvasive glucose sensing.

Types of Techniques	Advantages	Disadvantages
Optical (Direct)	Good correlation with blood glucose	Glucose resides in the skin Light cannot penetrate deeply into the skin Affected by interfering substances in the skin
Optical (Indirect)	Can be measured at the skin surface	Affected by many physiological and environmental factors
Transdermal	Easy to analyze after ISF is extracted from skin	Low glucose concentration in extracted ISF Cannot detect rapid changes due to long extraction process May cause discomfort to the user Affected by sweating
Electrical	Can probe the whole tissue	Weak correlation with blood glucose
Thermal	Easy to sense skin temperature	May not work for people without diabetes Affect by many physiological and environmental factors
Fusion	Multiple modalities can complement with each other	Additional hardware is required

**Table 5 sensors-23-07057-t005:** A summary of the in vivo studies performances grouped by the types of techniques. The studies are evaluated with the mean absolute relative error (MARD), Zones A and B of the Clarke’s error grid (or the consensus error grid if specified), correlation coefficient (r), coefficient of determination (R${}^{2}$), mean absolute difference (MAD), and root mean square error (RMSE).

Ref.	Year	Clinical Study			Study Result
		N	w/ Diabetes	w/o Diabetes	
NIR Spectroscopy					
[82]	2002	9	Yes	No	MARD: 20.6%; Zone A: 63.5%; Zone B: 34.9%
[84]	2010	36	No	Yes	r: 0.48; RMSE: 1.34 mmol/l; Zone A + B: 100.0%
[86]	2016	5	No	Yes	Zone A + B: 100.0%
[92]	2018	36	Yes	Yes	MARD: 14.4%; Zone A: 96.6%; Zone B: 3.4% (consensus)
[89]	2021	19	No	Yes	r: 0.92, Zone A: 97.96%
[79]	2022	635	Yes	Yes	Zone A: 100.0%
MIR Spectroscopy					
[85]	2014	3	No	Yes	Zone A: 84.0%
[87]	2018	6	No	Yes	r: 0.36; Zone A + B: 100.0%
Occlusion Spectroscopy					
[90]	2007	23	Yes	No	MARD: 17.2%; Zone A: 69.7%; Zone B: 25.7%
Photoacoustic Spectroscopy					
[98]	2015	30	No	Yes	MARD: 9.61% ± 10.55%. Zone A: 87.24%; Zone B: 12.76%
[99]	2017	24	No	Yes	MARD: 8.84%; Zone A: 92.86%; Zone B: 7.14%
[102]	2018	5	Yes	Yes	MAD: 16 ± 7 mg/dL.
Raman Spectroscopy					
[108]	2005	17	No	Yes	MARD: 7.8% ± 1.8%; R${}^{2}$: 0.83 ± 0.10
[109]	2009	30	Yes	No	MAD: 38 mg/dL; Zone A: 53.0%; Zone B: 39.0%
[110]	2018	35	Yes	No	MARD: 25.8%; Zone A + B: 93.0% (consensus)
[106]	2019	12	No	Yes	RMSEP = 0.27 mmol/L; R${}^{2}$ = 0.98; Zone A + B: 100.0%
[107]	2021	15	Yes	No	MARD: 26.3% ± 10.8%; Zone A + B: 93.6%
Polarimetry					
[123]	2020	50	Yes	Yes	MARD: 10.0%; Zone A: 89.0%; Zone B: 11.0%; r: 0.91
Photoplethysmography					
[136]	2019	30	Yes	Yes	r: 0.95
[137]	2019	611	Yes	Yes	Zone A: 80.6%; Zone B: 17.4%
[138]	2020	200	Yes	Yes	MARD: 7.62%
[139]	2020	8	Yes	Yes	r: 0.858; Zone A: 74.29%; Zone B: 25.71%
[140]	2021	26	n/a	n/a	Zone A: 96.15%; Zone B: 3.85%
Reverse Iontophoresis					
[147]	2001	231	Yes	Yes	MARD: 19.0%; r: 0.85; Zone A + B: 95.3%
[156]	2022	23	Yes	Yes	Zone A: 46.99%; Zone B: 37.35%
Metabolic Heat Conformation					
[186]	2004	10	Yes	Yes	r: 0.91
[187]	2017	31	Yes	Yes	r: 0.89; Zone A + B: 94.4%
Fusion Techniques					
[192]	2018	114	Yes	No	MARD: 22.7%; Zone A + B: 98.0%
[195]	2018	5	Yes	No	MAD: 3.794 mg/dL; r: 0.92; Zone A: 100.0%

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
