# Peer review of "Noninvasive Glucose Sensing In Vivo"

_sensors, 2023, doi:10.3390/s23167057_

Round 1

Reviewer 1 Report

Very detailed article describing all the most known methods.

The only thing I would like to see is a comparison table with the best data obtained for all methods. This will allow the reader and researchers to see the big picture of which method is currently the most studied.

Author Response

We thank the reviewer for the time to read this manuscript and the constructive comments. We have added a table (Table 5, P. 20) at the end of the methods section (Section 3) to show and compare the performance of the in vivo studies for all methods.

Reviewer 2 Report

This manuscript deals with an important and very interesting issue on the current advancements and progresses in noninvasive glucose sensing technology in vivo and offers indeed an insightful outlook on existing and future solutions. The language is in general correct and the references up to date. This comprehensive review differs from previous similar papers on this topic, focusing here specially on the in vivo studies demonstrating the feasibility of these technologies for human use. Despite significant and efficient advancements in technology, the remaining obstacles do not seem to allow a technique application generalizable across diabetic populations in the immediate future.

1. I'd like to see their opinion about the technical application that might be  generalized in diabetic populations in immediate future (if possible in 2-3 lines-sentences) 2. There are some similar papers in medline but this paper concerns in vivo applications. 3. This is in vivo and extended comprehensive review. 4. Add 1-2 sentences targeted on the methods immediately applicable in large populations. 5. Conclusions consistent and evident arguments but the addressed main question posed is on the techniques that might be used soon. 6. References are consistent, appropriate, and up to date. 7. Tables and figures satisfactory.

Author Response

We thank the reviewer for the time to read this manuscript and the constructive comments. We have revised the manuscript to address the reviewer’s concerns. Thank you for your time and expertise.

1. I'd like to see their opinion about the technical application that might be  generalized in diabetic populations in immediate future (if possible in 2-3 lines-sentences)
>> We have added our opinion on the technical application that could be generalized to the methods in the immediate future.

4. Add 1-2 sentences targeted on the methods immediately applicable in large populations.
>> There are some promising results in recent works, but there are still challenges to tackle before they can be immediately applicable. We have added some possible solutions to these challenges at the end of the discussion section.

Reviewer 3 Report

Overall, this is an interesting, well written, and very comprehensive review paper on the noninvasive glucose sensing technology in vivo. The manuscript could be made even stronger if the authors further improved the writing and also provided more context for the development and application of noninvasive glucose sensing technology. There are several issues which need some revision before publication. The following questions should be responded reasonably.

1. In the manuscript title, the authors mentioned the Closer Look. Could they discuss more on the Closer Look in the revised manuscript?

2. There are few figures in the manuscript. As a review paper, the typical figures should be provided.

3. There is the statement “Therefore, it is vital for PwD to monitor and regulate their blood glucose level regularly” in the Introduction section. I think that the reason is not enough. Besides, the authors didn't explain the meaning and purpose of review clearly in this section.

4. Table 4 showed the summary of optical techniques with in vivo study sorted according to year of publication. I think this section should be introduced in detail. Besides, the corresponding typical pictures should be added in the manuscript.

5. This manuscript is readable, but there are some grammar errors, which should be polished.

This manuscript is readable, but there are some grammar errors, which should be polished.

Author Response

We thank the reviewer for the time to read this manuscript and the constructive comments. We have revised the manuscript to address the reviewer’s concerns. Thank you for your time and expertise.

  1. In the manuscript title, the authors mentioned the Closer Look. Could they discuss more on the Closer Look in the revised manuscript?
    >> We have revised our manuscript title.
  2. There are few figures in the manuscript. As a review paper, the typical figures should be provided.
    >> We have added some figures to illustrate the basic setups for the methods with in vivo studies.
  3. There is the statement “Therefore, it is vital for PwD to monitor and regulate their blood glucose level regularly” in the Introduction section. I think that the reason is not enough. Besides, the authors didn't explain the meaning and purpose of review clearly in this section.
    >> We have added some possible complications caused by extreme blood glucose levels. We have also provided more information regarding diabetes and glucose management in the background section.
  4. Table 4 showed the summary of optical techniques with in vivo study sorted according to year of publication. I think this section should be introduced in detail. Besides, the corresponding typical pictures should be added in the manuscript.
    >> We have introduced the table in more detail at the beginning of section 3.1. The details of each work included in the table are subsequently explored in the following sections.
  5. This manuscript is readable, but there are some grammar errors, which should be polished.
    >>We have carefully checked for grammatical errors and further polished the manuscript.

Reviewer 4 Report

The manuscript is an excellent review, encompassing a wide range of noninvasive glucose sensing techniques while addressing the challenges and potential solutions for future research. I believe this review will be highly valuable for researchers seeking a comprehensive understanding of noninvasive glucose sensors. To further enhance the clarity and readability of the manuscript, I suggest including more schematics illustrating the various noninvasive sensing techniques. These schematics will help readers grasp the principles and differences of these techniques more effectively. Overall, this review is a valuable contribution to the field of noninvasive glucose sensing.

Author Response

We thank the reviewer for the time to read this manuscript and the constructive comments. We have revised the manuscript to address the reviewer’s concerns. In particular, we have added the schematics of the basic setups for the methods with in vivo studies to help explain the principles of these techniques to the readers. Thank you for your time and expertise.

Round 2

Reviewer 3 Report

I think the manuscript can be accepted.